# Beyond Correctness: Evaluating Subjective Writing Preferences Across Cultures

## Abstract

Current preference learning methods achieve high accuracy on standard benchmarks but exhibit significant performance degradation when objective quality signals are removed. We introduce **WritingPreferenceBench**, a dataset of 1,800 human-annotated preference pairs (1,200 English, 600 Chinese) across 8 creative writing genres, where responses are matched for objective correctness, factual accuracy, and length. On this benchmark, sequence-based reward models—the standard architecture for RLHF—achieve only 52.7% mean accuracy, while zero-shot language model judges perform at 53.9%. In contrast, generative reward models that produce explicit reasoning chains achieve 81.8% accuracy. We observe high within-model variance across genres: individual models range from 18.2% to 81.8% accuracy across different writing categories, with standard deviations averaging 10.1%. This variance persists regardless of model scale, with 27B parameter models showing no consistent improvement over 8B variants. Our results suggest that current RLHF methods primarily learn to detect objective errors rather than capture subjective quality preferences, and that successful preference modeling may require intermediate reasoning representations rather than direct classification. We release WritingPreferenceBench and human judge reuslts at `https://anonymous.4open.science/r/Writing-Preference-Bench`.

## 1 Introduction

Reinforcement learning from human feedback (RLHF) has become the dominant paradigm for aligning language models with human values (Christiano et al., 2017; Ouyang et al., 2022; Bai et al., 2022). Yet a troubling pattern emerges: models achieving 95% accuracy on RewardBench's objective tasks (Lambert et al., 2024) collapse to 51.2% on subjective writing preference when grammatical errors and factual mistakes are removed. This catastrophic degradation exposes a fundamental misalignment—current preference learning optimizes for error detection, not quality recognition.

However, writing tasks constitute over 40% of language model interactions (OpenAI, 2025; Anthropic, 2025b), spanning creative fiction, persuasive essays, and personal expression where subjective quality matters more than objective correctness. Yet our evaluation infrastructure remains anchored in verifiable metrics. RewardBench (Lambert et al., 2024) conflates safety with preference; WritingBench mixes creative with functional tasks (Wu et al., 2025); LitBench uses Reddit upvotes as quality proxies (Fein et al., 2025). Recent theoretical work warns of "reward hacking" where models exploit spurious correlations rather than learning genuine preferences (Pan et al., 2022)—but empirical evidence of this phenomenon in creative domains remains scarce.

We introduce WritingPreferenceBench, a benchmark that isolates subjective preference through systematic signal neutralization. Our dataset comprises 1,800 preference pairs (1,200 English, 600 Chinese) across 8 writing genres where both responses are grammatically correct, factually accurate, and length-matched. As Figure 1 shows, this controlled evaluation reveals catastrophic failures: sequence classifiers achieve near-random 53.6% accuracy, while all models exhibit genre instability ranging from 18% to 92%.

Our evaluation of 21 models uncovers systematic architectural failures. Sequence classifiers—the backbone of production RLHF systems (Rafailov et al., 2023)—average 52.7% accuracy, statistically indistinguishable from random. Language models as judges (Zheng et al., 2023) fare no better at 53.9%. Only generative reward models that produce explicit reasoning chains (Chen et al., 2025)

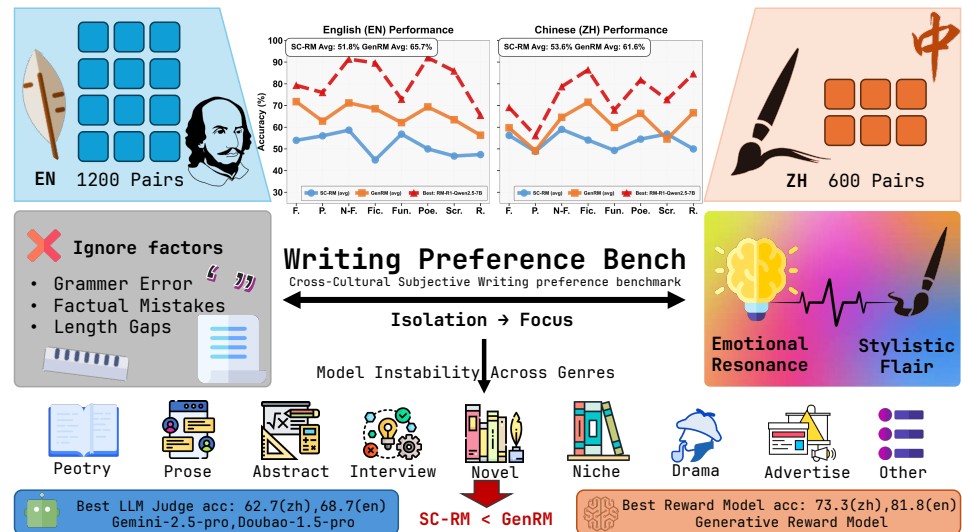

Figure 1: **WritingPreferenceBench** isolates subjective writing quality by neutralizing objective confounds (grammar, factuality, length). Across 1,800 human-validated preference pairs, standard sequence classifiers (SC-RM) perform near-randomly while generative reward models (GenRM) achieve 30% higher accuracy—but both architectures exhibit catastrophic instability across genres, exposing the brittleness of current preference learning.

show promise, with RM-R1-Qwen2.5-7B reaching 81.8%. This 30-point gap suggests that subjective preference requires intermediate representations—reasoning about quality rather than pattern matching against it.

The implications extend beyond evaluation. High within-model variance (standard deviation averaging 10.9% for sequence classifiers) reveals genre-specific overfitting: Skywork-Gemma-27B achieves 81.8% on Poetry but 40.4% on Humor. Scale provides no consistent benefit—the 27B model performs worse than 8B variants. Even reasoning-enhanced language models (Claude-4-Opus-thinking, OpenAI-o3) show no advantage over standard architectures, suggesting the limitation is representational rather than computational.

**Contributions.** We make three contributions to understanding preference learning:

- **Benchmark**: WritingPreferenceBench provides 1,800 validated preference pairs with systematic signal isolation, enabling reproducible evaluation of subjective preference across languages and genres.

- **Empirical findings**: Comprehensive evaluation establishes that (i) sequence classifiers fail systematically on subjective tasks, (ii) generative reward models with reasoning achieve 30% higher accuracy, and (iii) zero-shot LLM judges cannot reliably assess creative quality despite instruction tuning.

- **Architectural insights**: Evidence that successful preference learning requires intermediate reasoning representations, not just pattern matching, with implications for next-generation RLHF systems.

## 2 WRITINGPREFERENCEBENCH

The fundamental challenge in evaluating subjective writing is not merely collecting human judgments, but ensuring those judgments isolate genuine aesthetic and stylistic preference from objective quality signals. We present **WritingPreferenceBench**, a meticulously constructed benchmark that addresses this challenge through 1,800 preference pairs spanning English and Chinese creative writing. The construction process, illustrated in Figure 2, was guided by rigorous design principles and implemented through a human-in-the-loop pipeline designed to systematically eliminate confounding variables.

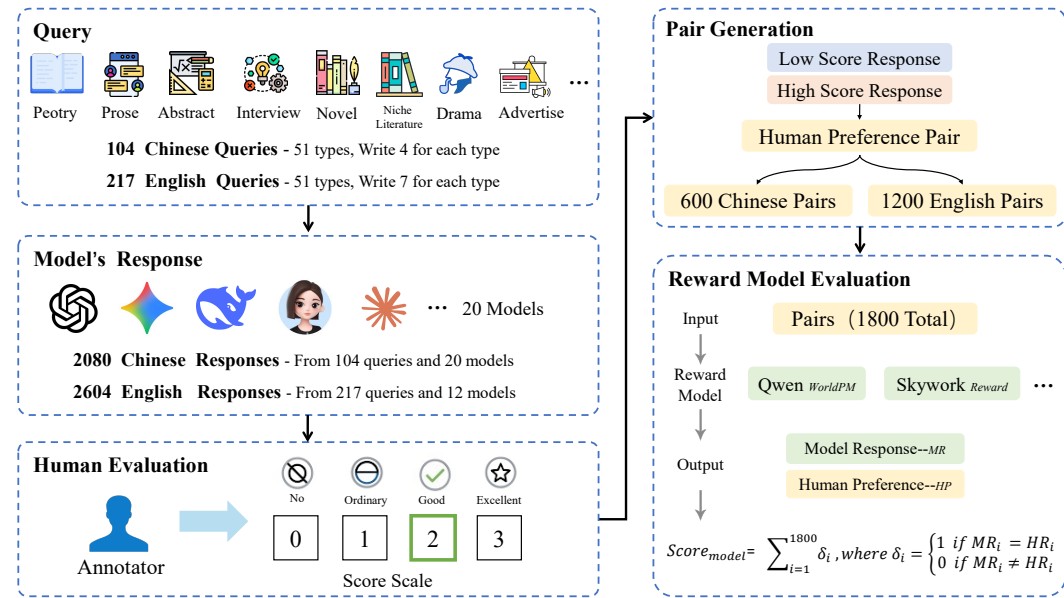

Figure 2: The data curation pipeline of **WritingPreferenceBench**. Our multi-stage process begins with expert-crafted queries across 51 genres, generates diverse responses using 20 state-of-the-art models, and culminates in rigorous human evaluation by trained annotators. Quality control mechanisms operate throughout to ensure preference pairs reflect genuine subjective quality distinctions rather than objective differences.

## 2.1 BENCHMARK CONSTRUCTION PIPELINE

We implemented a multi-stage pipeline that translates our design principles into a concrete, reproducible workflow. This process, depicted in Figure 2, first generates a diverse and culturally-rich set of candidate responses and then applies a rigorous, human-led filtering protocol to isolate pairs that represent genuine subjective preference.

**Phase 1: Architecting Diverse and Culturally-Rich Queries.** The benchmark's foundation is a taxonomy of 51 creative writing categories, developed by merging taxonomies from established writing communities. To ensure both representational diversity and practical relevance, these categories range from classical literary traditions to contemporary forms like advertising copy. Query development followed a dual-expertise workflow where two veteran creative writing instructors drafted and aligned on creative blueprints for each category. A leading instruction-tuned LLM then expanded these blueprints into full queries, which were validated by the experts for creative intent, cultural neutrality, and evaluative granularity through 3–5 rounds of refinement.

**Phase 2: Generating a Spectrum of Responses.** To create a rich and varied corpus for curation, we utilized a diverse suite of 20 state-of-the-art language models, including GPT-4.1, Claude-4, Gemini-2.5-Pro, and Doubao-1.5-Pro. For each query, every model produced 5 outputs with temperature sampling set to $T = 0.8$. This strategy ensured the generation of a wide spectrum of quality—from formulaic to highly original—providing the necessary variance to identify the controlled preference gaps central to our benchmark's design.

**Phase 3: Human-in-the-Loop Annotation and Quality Control.** This phase is the cornerstone of our methodology, operationalizing a focus on subjective quality through a rigorous, integrated annotation and filtering protocol.

- **Initial Triage: Filtering for Objective Correctness.** Before subjective assessment, an automated screening process removed responses with objective deficiencies. This filter eliminated approximately 15% of the raw responses, discarding outputs with comprehension-impeding grammatical errors, factual inconsistencies, or clear prompt violations. This crucial step ensures our benchmark tests for subjective quality, not basic error detection.

- **Expert Evaluation with a Calibrated Rubric.** We recruited 7 expert annotators with demonstrated writing proficiency. After a calibration phase on 50 consensus-scored examples, annotators independently scored responses on a 4-point scale designed to distinguish levels of creative quality:

  - **3 (Creative):** Demonstrates genuine creativity, stylistic flair, and deep engagement; suitable for publication.
  - **2 (Competent):** Well-structured and complete, but predictable and lacking in originality.
  - **1 (Formulaic):** Technically correct but lacks creative engagement; follows a rigid template.
  - **0 (Elementary):** Fundamentally flawed or off-topic.

  The framework includes universal quality criteria and detailed genre-specific standards (see Appendix C for complete guidelines).

- **Statistical Validation and Final Pair Curation.** A final preference pair was curated only if it met three strict criteria for reliability and validity. A pair was accepted only if:

  1. It had **directional agreement** from at least 2 of the 3 annotators.
  2. It showed a **minimum score gap** of $\Delta \geq 1$.
  3. It passed a check for **absence of confounding factors**, such as significant length disparities.

  Finally, a separate team of bilingual experts performed a cross-cultural validation on a sample of pairs to confirm that scoring standards were applied equivalently across both languages.

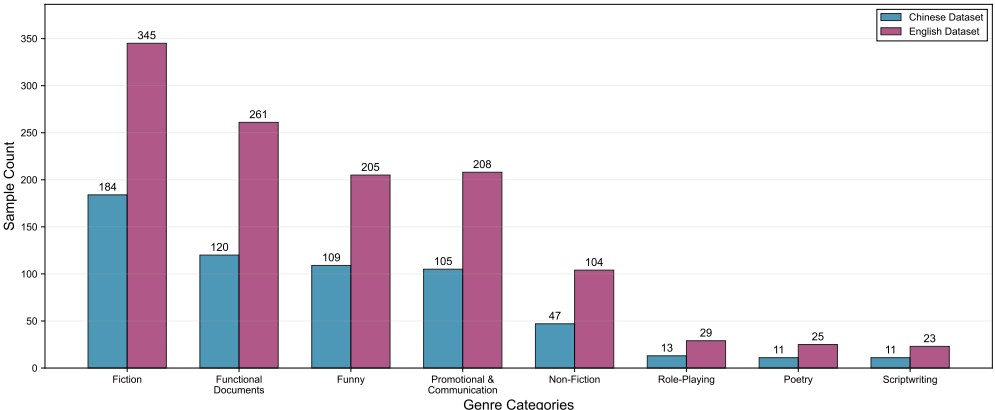

Figure 3: Distribution of preference pairs across a sample of the 8 writing macro-categories for both English and Chinese in **WritingPreferenceBench**. The dataset maintains balanced coverage across diverse writing genres, with deliberate oversampling of underrepresented categories to ensure comprehensive evaluation of preference modeling capabilities.

## 2.2 DATASET STATISTICS

**WritingPreferenceBench** comprises 1,800 human-validated preference pairs (1,200 English, 600 Chinese) that establish a new standard for evaluating subjective writing preferences. Unlike existing benchmarks that conflate objective correctness with aesthetic quality, our dataset isolates genuine stylistic preference through careful statistical design.

**Compositional Structure and Coverage.**

Figure 3 reveals the deliberate architectural choices underlying our 51-category taxonomy. The dataset's statistical properties reflect three critical design decisions:

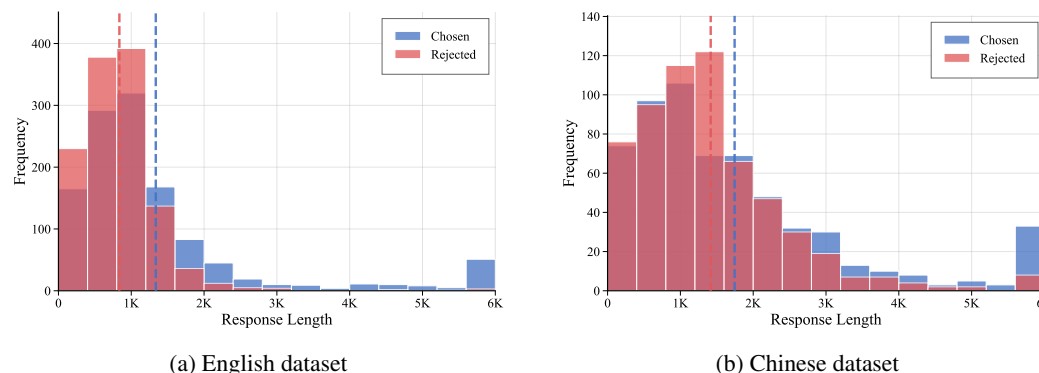

(a) English dataset                    (b) Chinese dataset

Figure 4: Length distributions of chosen and rejected responses reveal the statistical signature of creative quality in **WritingPreferenceBench**. Distributions truncated at 6K for visualization; dashed lines indicate means.

- **Cross-lingual parity**: While maintaining a 2:1 English-Chinese ratio due to annotator availability, we ensure equivalent statistical power across languages with a minimum of 20 pairs per category in each language.
- **Genre equilibrium**: Each category contains 20-40 preference pairs (mean=35.3, std=7.2), a distribution engineered to prevent the genre collapse observed in web-scraped datasets.
- **Compositional diversity**: The taxonomy spans 8 macro-categories with deliberate over-sampling of traditionally underrepresented genres (e.g., poetry, scriptwriting) to stress-test models' preference modeling capabilities beyond dominant web text distributions.

**Statistical Validation of Subjective Quality Gaps.** Figure 4 and Table 3 reveal the empirical signature of subjective preference in our dataset. The length distributions expose a critical phenomenon: chosen responses exhibit significantly higher variance (English: SD=1801.9 vs. 593.4; Chinese: SD=1967.5 vs. 1311.0) and right-skewness compared to rejected responses. This asymmetry reflects a fundamental property of creative excellence—while mediocrity converges toward formulaic patterns, creativity manifests across diverse scales. The score distributions validate our annotation protocol's effectiveness. The median scores (English: 3 vs. 2; Chinese: 3 vs. 1) align precisely with our rubric's creative-competent and creative-formulaic boundaries, demonstrating that our benchmark captures the most informative preference contrasts.

## 3 EXPERIMENTS

We evaluate 21 models on WritingPreferenceBench: 7 reward models and 14 language models serving as zero-shot judges. This section describes our evaluation protocols and experimental setup.

### 3.1 EVALUATION PROTOCOLS

**Protocol 1: Reward Model Scoring.** For each preference pair $(R_{\text{chosen}}, R_{\text{rejected}})$, reward models assign scalar scores. A prediction is correct if $\text{RM}(R_{\text{chosen}}) > \text{RM}(R_{\text{rejected}})$. We compute accuracy as:

$$\text{Accuracy} = \frac{1}{N} \sum_{i=1}^{N} \mathbb{I}[\text{RM}(R_{\text{chosen}}^{(i)}) > \text{RM}(R_{\text{rejected}}^{(i)})]$$

where $N$ denotes total preference pairs and $\mathbb{I}[\cdot]$ is the indicator function.

**Protocol 2: Pairwise Preference Judgment.** Language models receive both responses with instructions to select the preferred text based on creativity, style, and emotional resonance. We use deterministic decoding $(T = 0)$ and extract preferences from model outputs. This protocol tests whether general-purpose models can perform zero-shot preference evaluation without specialized training.

## 3.2 MODELS

**Reward Models.** We evaluate 7 models spanning different architectures and scales:

- **Sequence Classifiers**: Nvidia/AceMath-7B-RM (Liu et al., 2024b), RM-Mistral-7B (Dong et al., 2023), Skywork-Reward-Llama-3.1-8B (Liu et al., 2024a), Skywork-Reward-Gemma-2-27B Liu et al. (2024a)
- **Generative RMs**: RM-R1-DeepSeek-Qwen-7B, RM-R1-DeepSeek-Qwen-14B, RM-R1-Qwen2.5-7B (Chen et al., 2025; Team, 2024b; Guo et al., 2025)

**Language Model Judges.** We evaluate 14 models including reasoning-enhanced variants (Claude-4-{Opus, Sonnet}-thinking (Anthropic, 2025a), Doubao-1.6-thinking (Seed et al., 2025), OpenAI-o3-high (OpenAI, 2025)) and standard models (Gemini-2.5-{Flash, Pro} (Team, 2024a), DeepSeek-R1 (Guo et al., 2025), ByteDance-Seed-1.6 (Team, 2025), Doubao-{1.5-Lite, 1.5-Pro, 1.6-flash} (ByteDance Seed, 2025), Qwen-3-235B (Team, 2024b), OpenAI-o4-mini).

## 3.3 IMPLEMENTATION DETAILS

All experiments use the same prompt templates across models to ensure fair comparison. For reward models, we use the default inference configuration from their respective repositories. For LLM judges, we employ a standardized prompt format that presents both responses and requests a preference judgment with justification. We evaluate on the full WritingPreferenceBench dataset (1,200 English, 600 Chinese pairs) without subsampling.

## 4 RESULTS

## 4.1 REWARD MODEL PERFORMANCE

We evaluate seven reward models across WritingPreferenceBench. Following RewardBench (Lambert et al., 2024), models divide into sequence classifiers (discriminative heads on language models) and generative reward models. However, our results reveal that the RM-R1 series (Chen et al., 2025) represents a distinct category—generative models that produce reasoning chains before preference judgments, diverging from both traditional classifiers and DPO-based approaches evaluated in RewardBench. Table 1 presents accuracy across eight writing genres in English and Chinese.

**Generative architecture enables strong performance.** Generative reward models achieve substantially higher accuracy than sequence classifiers. RM-R1-Qwen2.5-7B reaches 81.8% (EN)—the highest performance across all models and 30 percentage points above the sequence classifier mean. All three generative models exceed 50% accuracy, while three of four sequence classifiers fall below random chance. This architectural advantage persists across languages: best generative performance reaches 64.5% (ZH) versus 53.5% for sequence classifiers.

**Scale improves stability, not just accuracy.** Scaling from 7B to 14B in RM-R1-DeepSeek yields distinct benefits: accuracy improves (50.3%→62.6% ZH), but more critically, variance drops (9.8→5.5). This stability gain does not transfer to sequence classifiers—Skywork-Gemma-27B shows no improvement over 8B variants despite 3.4× parameters. The 14B model's low variance (5.5) represents the most consistent performance across genres, suggesting scale enables robust preference representations in generative architectures.

**Sequence classifiers exhibit catastrophic genre failure.** All sequence classifiers demonstrate extreme performance swings: Nvidia/AceMath-7B ranges from 18.2% to 61.5% (43.3 percentage point gap), while Skywork-Gemma-27B varies from 21.7% to 81.8%. Mean within-model standard deviation reaches 10.1% for discriminative models versus 9.2% for generative. These genre-specific failures—often below 40% accuracy—indicate fundamental instability rather than minor variations.

**Cross-lingual consistency reveals architectural robustness.** Generative models maintain more consistent cross-lingual performance than sequence classifiers. RM-R1-DeepSeek-Qwen-14B achieves 62.6% (ZH) and 62.5% (EN), while sequence classifiers show larger gaps:

Table 1: Reward model accuracy (%) on **WritingPreferenceBench** by architecture. Colors: >70% , <50% . Best overall in **bold**.

| Model | Lang | Func. | Promo. | Non-Fic. | Fiction | Funny | Poetry | Script | Role | Avg | Std |
|---|---|---|---|---|---|---|---|---|---|---|---|
| *Sequence Classifiers (Discriminative with scalar output)* | | | | | | | | | | | |
| Nvidia/AceMath-7B | ZH | 56.7 | 50.5 | 55.3 | 54.9 | 52.3 | 18.2 | 54.6 | 61.5 | 53.5 | 11.2 |
| | EN | 48.0 | 53.9 | 59.6 | 33.9 | 55.1 | 36.0 | 21.7 | 51.7 | 46.8 | 12.4 |
| RM-Mistral-7B | ZH | 60.8 | 46.7 | 63.8 | 55.9 | 54.1 | 54.5 | 72.7 | 46.1 | 55.6 | 9.1 |
| | EN | 65.2 | 60.0 | 54.8 | 62.9 | 64.9 | 72.0 | 78.3 | 44.8 | 62.6 | 10.1 |
| Skywork-Llama-3.1-8B | ZH | 56.7 | 43.8 | 61.7 | 52.2 | 50.5 | 63.6 | 54.6 | 38.5 | 52.0 | 8.2 |
| | EN | 53.6 | 56.3 | 60.6 | 49.0 | 52.2 | 56.0 | 65.2 | 41.4 | 53.1 | 7.3 |
| Skywork-Gemma-2-27B | ZH | 50.8 | 54.3 | 55.3 | 53.3 | 40.4 | 81.8 | 45.5 | 53.9 | 51.2 | 11.3 |
| | EN | 49.0 | 53.9 | 59.6 | 33.9 | 55.1 | 36.0 | 21.7 | 51.7 | 46.8 | 12.4 |
| *Generative Reward Models (Reasoning before scoring)* | | | | | | | | | | | |
| RM-R1-DeepSeek-Qwen-7B | ZH | 50.8 | 45.7 | 57.5 | 56.5 | 45.0 | 27.3 | 36.4 | 46.2 | 50.3 | 9.8 |
| | EN | 64.8 | 50.9 | 58.7 | 57.4 | 54.4 | 52.0 | 52.2 | 37.9 | 56.8 | 7.2 |
| RM-R1-DeepSeek-Qwen-14B | ZH | 59.2 | 45.7 | 57.5 | 71.7 | 66.7 | 90.0 | 54.6 | 69.2 | 62.6 | 13.2 |
| | EN | 71.3 | 61.5 | 63.5 | 58.6 | 59.0 | 64.0 | 52.2 | 65.5 | 62.5 | 5.5 |
| RM-R1-Qwen2.5-7B | ZH | 69.1 | 56.1 | 78.7 | 86.4 | 67.9 | 81.8 | 72.7 | 84.6 | **73.3** | 10.9 |
| | EN | 79.3 | 76.0 | 91.4 | 89.6 | 72.9 | 92.0 | 85.9 | 65.5 | **81.8** | 9.5 |

Nvidia/AceMath-7B scores 53.5% (ZH) versus 46.8% (EN). This consistency in generative models, particularly at larger scales, suggests that reasoning-based architectures learn more language-agnostic preference representations.

## 4.2 LANGUAGE MODEL JUDGE PERFORMANCE

Table 2 presents the performance of 14 state-of-the-art language models serving as zero-shot preference judges on WritingPreferenceBench, revealing systematic underperformance compared to specialized reward models.

**LLM judges systematically underperform reward models.** General-purpose language models achieve mean accuracy of 53.9%, compared to 58.2% for reward models—a 4.3% degradation despite orders of magnitude more parameters. The best LLM judge (Doubao-1.5-Pro: 68.7% EN) remains 13.1% below the top generative reward model (RM-R1-Qwen2.5-7B: 81.8% EN). This gap persists across all model families and scales, indicating that zero-shot preference evaluation cannot match task-specific training.

**Reasoning capabilities provide no systematic advantage.** Models with explicit reasoning mechanisms show no consistent improvement over standard architectures. Claude-4-Opus-thinking achieves 61.0% (EN) while non-reasoning Doubao-1.5-Pro reaches 68.7%. Similarly, OpenAI-o3-high with advanced reasoning scores only 48.1%, performing worse than simpler models like Gemini-2.5-Flash (57.5%). The correlation between reasoning capability and preference accuracy is negligible (r=0.08, p¿0.5), suggesting that chain-of-thought processing does not inherently improve subjective quality assessment.

**Genre instability exceeds that of reward models.** LLM judges exhibit extreme performance variance across genres, surpassing even sequence classifiers. Gemini-2.5-Pro ranges from 80.0% on English Poetry to 34.8% on Scriptwriting—a 45.2% gap. OpenAI-o3-high shows similar instability: 72.0% on Poetry versus 21.7% on Scriptwriting. Mean within-model standard deviation reaches 11.4%, with 9 of 14 models showing standard deviations exceeding 10%. This variance pattern suggests that LLMs rely on superficial genre markers rather than genuine quality assessment.

**Cross-lingual performance reveals model-specific biases.** LLM judges demonstrate inconsistent cross-lingual patterns. Doubao models maintain relative consistency (1.5-Pro: 62.5% ZH, 68.7% EN), while others show severe degradation: OpenAI-o3-high drops from 48.1% (EN) to 42.0% (ZH). These disparities do not correlate with known multilingual capabilities, suggesting that preference evaluation activates different model behaviors across languages.

**Implications for LLM-as-judge paradigm.** The systematic underperformance of LLM judges relative to specialized reward models challenges the widespread adoption of LLM-as-judge evalua-

Table 2: Language model judge accuracy (%) on **WritingPreferenceBench** using pairwise preference evaluation. Colors: >70% , <50% . Best overall in **bold**.

| Model | Lang | Func. | Promo. | Non-Fic. | Fiction | Funny | Poetry | Script | Role | Avg | Std |
|---|---|---|---|---|---|---|---|---|---|---|---|
| ByteDance-Seed-1.6 | ZH | 42.1 | 32.2 | 52.5 | 59.9 | 38.5 | 45.5 | 36.4 | 38.5 | 45.5 | 9.4 |
| | EN | 54.6 | 54.2 | 49.2 | 41.1 | 47.3 | 68.0 | 17.4 | 41.4 | 48.3 | 13.4 |
| Claude-4-Opus-thinking | ZH | 55.1 | 36.4 | 57.6 | 73.3 | 49.5 | 54.6 | 72.7 | 46.2 | 56.0 | 12.1 |
| | EN | 65.7 | 64.3 | 64.1 | 60.1 | 54.2 | 64.0 | 43.5 | 51.7 | 61.0 | 7.3 |
| Claude-4-Sonnet-thinking | ZH | 46.7 | 38.1 | 62.7 | 65.7 | 48.6 | 54.6 | 63.6 | 46.2 | 52.8 | 9.9 |
| | EN | 62.4 | 58.6 | 58.7 | 53.9 | 50.3 | 50.0 | 31.8 | 48.2 | 55.7 | 9.3 |
| DeepSeek-R1 | ZH | 46.7 | 41.5 | 61.0 | 61.6 | 48.6 | 45.5 | 63.6 | 46.2 | 52.0 | 8.8 |
| | EN | 57.4 | 61.7 | 43.8 | 40.8 | 42.9 | 72.0 | 17.4 | 51.7 | 49.3 | 15.4 |
| Doubao-1.5-Lite | ZH | 44.9 | 42.4 | 64.4 | 62.8 | 44.0 | 36.4 | 72.7 | 38.5 | 51.5 | 13.0 |
| | EN | 62.8 | 63.9 | 42.2 | 50.5 | 45.4 | 52.0 | 47.8 | 48.3 | 53.7 | 8.1 |
| Doubao-1.5-Pro | ZH | 54.2 | 47.5 | 72.9 | 79.7 | 54.1 | 54.6 | 63.6 | 69.2 | 62.5 | 10.8 |
| | EN | 74.4 | 74.5 | 64.8 | 71.0 | 57.1 | 68.0 | 56.5 | 58.6 | **68.7** | 7.2 |
| Doubao-1.6-flash | ZH | 39.3 | 30.5 | 55.9 | 60.5 | 38.5 | 54.6 | 63.6 | 38.5 | 45.8 | 11.9 |
| | EN | 57.0 | 53.7 | 49.2 | 43.6 | 46.3 | 52.0 | 30.4 | 44.8 | 49.3 | 7.7 |
| Doubao-1.6-thinking | ZH | 46.7 | 35.6 | 66.1 | 73.8 | 43.1 | 72.7 | 54.6 | 46.2 | 54.2 | 13.9 |
| | EN | 64.1 | 66.5 | 60.9 | 57.9 | 48.8 | 72.0 | 30.4 | 41.4 | 58.9 | 12.8 |
| Doubao-1.6-thinking-agent | ZH | 48.6 | 40.7 | 66.1 | 71.5 | 49.5 | 63.6 | 63.6 | 53.9 | 56.2 | 10.3 |
| | EN | 64.9 | 61.7 | 61.7 | 55.5 | 47.3 | 68.0 | 39.1 | 48.3 | 57.6 | 9.6 |
| Gemini-2.5-Flash | ZH | 47.7 | 34.8 | 61.0 | 68.0 | 45.0 | 63.6 | 63.6 | 38.5 | 52.2 | 12.1 |
| | EN | 59.1 | 57.7 | 62.5 | 59.8 | 52.2 | 56.0 | 34.8 | 51.7 | 57.5 | 8.1 |
| Gemini-2.5-Pro | ZH | 53.3 | 44.9 | 71.2 | **80.2** | **56.9** | **72.7** | 72.7 | 61.5 | **62.7** | 11.3 |
| | EN | 70.3 | 66.5 | **68.0** | 65.7 | **58.5** | **80.0** | 34.8 | **72.4** | 65.7 | 12.6 |
| OpenAI-o4-mini | ZH | 43.0 | 33.9 | 54.2 | 50.6 | 35.8 | 45.5 | 63.6 | 38.5 | 43.5 | 9.9 |
| | EN | 58.3 | 58.6 | 60.9 | 55.5 | 53.2 | 68.0 | 30.4 | 55.2 | 56.6 | 10.0 |
| OpenAI-o3-high | ZH | 36.5 | 28.0 | 52.5 | 50.6 | 40.4 | 63.6 | 54.6 | 38.5 | 42.0 | 11.1 |
| | EN | 55.4 | 45.8 | 54.7 | 46.7 | 41.0 | 72.0 | 21.7 | 41.4 | 48.1 | 14.0 |
| Qwen-3-235B | ZH | 45.8 | 39.0 | 59.3 | 62.2 | 42.2 | 54.6 | 63.6 | 53.9 | 50.5 | 9.2 |
| | EN | 57.9 | 46.7 | 45.3 | 43.0 | 40.5 | 64.0 | 17.4 | 41.4 | 46.4 | 13.3 |

tion (Zheng et al., 2023). Mean accuracy of 53.9%—barely above random—indicates that zero-shot prompting cannot elicit reliable preference judgments for subjective tasks. The failure of reasoning-enhanced models further suggests that the limitation is not computational but representational: without explicit preference training, even advanced LLMs default to surface-level heuristics rather than genuine quality assessment.

## 5 DISCUSSION

Our findings reveal fundamental limitations in current preference learning paradigms when applied to subjective domains and expose a critical gap between model capabilities and genuine human aesthetic judgment.

**A Performance Ceiling on Subjectivity.** Even the best-performing models struggle to surpass a modest accuracy threshold on purely subjective tasks. The top bilingual reward model achieves only 62.5% accuracy, suggesting that current methods are more adept at identifying objective flaws (which we filtered out) than they are at capturing nuanced stylistic and creative preferences. This indicates a potential ceiling for architectures trained primarily on objective-centric data.

**Explicit Reasoning is No Panacea.** The systematic underperformance of advanced LLM judges compared to simpler, specialized reward models is a striking result. It challenges the prevailing assumption that enhanced reasoning capabilities, such as chain-of-thought, naturally lead to better alignment with complex human values. Our results suggest the problem is not one of logic but of representation; models lack the underlying framework to encode and weigh aesthetic qualities like originality, emotional resonance, or stylistic flair.

**Preference Functions are Brittle and Unstable.** The most concerning discovery is the extreme performance variance of individual models across genres. Swings of over 50 percentage points between categories like Poetry and Scriptwriting reveal that models are not learning generalizable

principles of "good writing." Instead, they appear to be memorizing brittle, genre-specific heuristics. This instability has profound implications for RLHF, as optimizing against such a volatile reward signal could introduce unpredictable and undesirable biases into model behavior, rewarding stylistic mimicry over genuine quality.

**No Evidence of Principled Cultural Understanding.** Our cross-lingual analysis found no consistent cultural or linguistic bias across all models. Instead, performance differences between English and Chinese appear to be model-specific artifacts, likely stemming from imbalances in training corpora. This suggests that current models do not possess a principled understanding of cultural nuance but rather reflect the idiosyncratic cultural footprint of their training data. Achieving true cross-cultural alignment will require more than simply adding multilingual data; it necessitates fundamentally new approaches that can model cultural context explicitly.

## 6 RELATED WORK

**Preference Learning and Evaluation Benchmarks.** Modern preference learning originated with Christiano et al. (Christiano et al., 2017), scaled through InstructGPT (Ouyang et al., 2022) and RLHF (Christiano et al., 2017). Subsequent benchmarks (Gao et al., 2023; Bai et al., 2022; Rafailov et al., 2023; Chiang et al., 2024) and comprehensive evaluations like RewardBench (Lambert et al., 2024), AlpacaEval (Li et al., 2023), and MT-Bench (Zheng et al., 2023) measure conversation, reasoning, and instruction-following. However, these benchmarks conflate objective correctness with subjective preference. RewardBench achieves 95% on safety but cannot evaluate aesthetic judgment; MT-Bench measures factual accuracy, not creative quality. Our work reveals that models excelling on these benchmarks fail catastrophically (52.7% accuracy) when objective signals are neutralized.

**Evaluating Creative and Subjective Writing.** Creative writing evaluation faces inherent subjectivity challenges (Chodorow et al., 2007; Burstein et al., 2003; Miltsakaki & Kukich, 2000; Li et al., 2018; Tay et al., 2018). Early reference-based metrics (BLEU, ROUGE) fail for open-ended generation. Recent benchmarks make progress but retain critical limitations. LitBench (Fein et al., 2025) uses Reddit upvotes—confounding preference with popularity and timing—and covers only English. WritingBench (Wu et al., 2025) spans six domains but mixes subjective creative tasks with objective functional ones (Academic & Engineering, Finance & Business). AlignBench (Liu et al., 2023) evaluates Chinese LLM alignment but focuses on general capabilities rather than creative preference. WritingPreferenceBench advances beyond these by: (1) systematically neutralizing objective confounds through human validation, (2) providing cross-lingual coverage with consistent methodology, and (3) isolating purely subjective quality discrimination where prior benchmarks conflate multiple signals.

## 7 CONCLUSION

We introduced **WritingPreferenceBench**, a benchmark isolating subjective writing preference through systematic neutralization of objective quality signals. Our empirical evaluation demonstrates that sequence-based reward models—the dominant RLHF architecture—achieve 52.7% accuracy on subjective preference tasks. In contrast, generative reward models incorporating explicit reasoning achieve 81.8% accuracy, suggesting that intermediate representations are necessary for subjective quality assessment. All evaluated models exhibit high variance across genres ($\sigma$=10.1-14.0%), with individual models ranging from 18.2% to 92% accuracy across categories, indicating reliance on genre-specific heuristics rather than generalizable preference functions.

These results have theoretical and practical implications for preference learning. The 30 percentage point performance gap between architectures challenges the direct preference optimization paradigm (Rafailov et al., 2023) and suggests that subjective domains require fundamentally different inductive biases than objective tasks. The failure of scale to improve performance (27B models underperform 7B variants) and the inability of reasoning-enhanced LLMs to surpass task-specific training indicate that current scaling laws may not apply to subjective preference modeling. Future work should investigate hybrid architectures combining the computational efficiency of discriminative models with the representational capacity of generative reasoning, and develop training objectives that explicitly encourage genre-invariant preference learning.

ETHICS STATEMENT

The authors of this work have read and commit to adhering to the ICLR Code of Ethics. This research involves the collection of subjective data from human participants and the creation of a public dataset, raising several ethical considerations that we have worked to address responsibly.

- **Human Subjects:** Our benchmark, WritingPreferenceBench, was constructed using judgments from 7 expert human annotators. All participants were recruited based on demonstrated professional expertise in creative writing. They were informed of the research goals, provided informed consent prior to participation, and were compensated for their skilled labor at a rate significantly exceeding the local minimum wage. All data collected was fully anonymized to protect participant privacy.

- **Data and Bias:** The dataset captures subjective human preferences, which are inherently influenced by cultural and individual backgrounds. To mitigate sampling bias, we recruited annotators with expertise in both English and Chinese literary contexts and used a detailed, calibrated scoring rubric to standardize evaluations (see Appendix C). However, we acknowledge that the preferences captured by our small group of experts may not generalize to all cultural or demographic groups. We encourage users of the benchmark to be mindful of its scope and limitations.

- **Potential Misuse:** We intend for this benchmark to be used to diagnose weaknesses in current preference learning models and spur research into more robust, nuanced architectures. We caution against interpreting its results as a universal standard for "good" writing or using it to train models that enforce a single, homogenous creative style, which could stifle creative diversity.

REPRODUCIBILITY STATEMENT

To ensure the reproducibility and extensibility of our work, we have made all relevant artifacts publicly available.

- **Dataset:** The complete **WritingPreferenceBench** dataset, containing all 1,800 human-validated preference pairs with associated metadata, is available.

- **Code:** Our evaluation scripts used to generate the results for all reward models and LLM judges (Tables 1 and 2) are included.

- **Documentation:** Detailed descriptions of our data curation pipeline, expert annotator guidelines, and scoring rubrics are provided in Section 2 and Appendix C. The exact models and prompts used in our experiments are detailed in Section 4.

All materials are accessible for review at the following anonymized URL: `https://anonymous.4open.science/r/Writing-Preference-Bench`.

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

## A  USE OF LARGE LANGUAGE MODELS

In accordance with ICLR 2026 policies, we disclose that a large language model was used during the manuscript preparation process to polish and refine the text. The LLM assisted in improving sentence fluency, enhancing clarity of expression, and standardizing language to align with academic writing conventions. All original academic arguments, experimental design, data analysis, and logical structure were developed solely by the authors. The authors independently verified all factual claims and technical content, and take full responsibility for the accuracy and validity of all statements in this paper.

## B  FULL TAXONOMY OF WRITING CATEGORIES

The benchmark spans 51 distinct writing categories, which are grouped into the 8 high-level domains shown below. This comprehensive taxonomy ensures a diverse and representative evaluation of model capabilities across a wide spectrum of writing tasks.

**Functional Documents**

- Abstract for Academic Paper
- Experiment Report
- Meeting Minutes
- Resume / Cover Letter
- Thank You / Apology Letter
- Product Manual
- Proposal / Plan
- Interview Questions
- Open Letter
- Argumentative Essay
- Eulogy / Memorial Text

**Promotional & Communication Documents**

- Speech Transcript
- Advertisement Copy / Marketing Email
- Slogan / Tagline
- Social Media Content
- Blog Post
- Product Review
- Popular Science Article
- Tutorial / Guide
- Debate Script

**Non-Fiction Writing**

- Prose / Essay
- Biography
- Travelogue
- Book / Film / Music Review

**Fiction**

- Fantasy / Magic
- Science Fiction
- Suspense / Mystery
- Historical Story
- Fairy Tale / Fable
- Slice of Life Story
- Emotional / Romance Story

- Wuxia
- Military Fiction
- Historical Fiction (Costume)
- Xuanhuan
- Xianxia
- Gaming Fiction
- Sports Fiction
- General Fiction / Story

**Funny**

- ACGN Funny Literature / Doujin (Fan Fiction)
- Fandom Funny Literature
- Esports / Gaming Funny Literature
- Hip-hop / Rap Culture Funny Literature
- Internet Slang Systems
- Anti-Mainstream Consumer Culture
- Cross-national / Cross-lingual Funny Literature
- Subculture Identity Expression
- Role-Playing (as a sub-genre)
- Funny Literature / Subculture

**Poetry**

- Poetry

**Scriptwriting**

- Play / Script

**Role-Playing**

- Role-Playing

## C  GENRE-SPECIFIC SCORING GUIDELINES

This appendix details the comprehensive framework provided to our expert annotators for evaluating model responses. The process is designed to be rigorous and consistent, combining a general quality rubric with a detailed hierarchy of universal and genre-specific standards.

### C.1  GENERAL SCORING RUBRIC (4-POINT SCALE)

Each response was assigned a holistic quality score from 0 to 3. The rubric was anchored with descriptive levels and analogies to everyday standards to ensure annotator calibration.

- **Score 3: Creative / Professional** The response is creative, stylistically fluent, and feels natural. It is a complete, well-crafted article on par with professionally published work (e.g., in a literary magazine). It is original, engaging, and often exceeds the prompt's expectations in a surprising way.
- **Score 2: Competent / Predictable** The response is good overall but lacks originality. The structure is sound and the content addresses the prompt, but the narrative or arguments are predictable. This level is analogous to a well-written but standard university-level essay or a competent product manual.
- **Score 1: Formulaic / Flawed** The response exhibits significant issues. It may be written in a language different from the one requested, or it follows a rigid, unnatural template (e.g., every paragraph starting with a subheading). The word choice can be awkward or inappropriately technical (e.g., using "quantum" in a non-scientific context). This is comparable to a middle-school-level essay.
- **Score 0: Incoherent / Irrelevant** The response is fundamentally unusable. It is nonsensical, completely fails to address the prompt's genre or topic, or consists mostly of a direct repetition of the query. This is analogous to an elementary-school or illiterate level of writing.

## C.2 UNIVERSAL EVALUATION CRITERIA

Beyond the holistic score, annotators assessed responses against a set of universal criteria applicable to all forms of writing.

- **Prompt Adherence and Intent:**
  - Does the response satisfy all explicit constraints in the query (e.g., themes, content, word count)?
  - Does it avoid vague, grandiose statements and focus on the core task?
  - Is the overall reading experience fluent, not sacrificed for overly ornate or complex sentences?
- **Structure and Coherence:**
  - Is the overall structure complete and are paragraphs divided logically?
  - Is the line of reasoning clear and are the ideas logically self-consistent?
  - For narratives, is the pacing effective (i.e., a clear beginning, development, climax, and conclusion)?
- **Content and Substance:**
  - Is the content rich and specific, avoiding empty, generic statements?
  - Does the chosen material effectively support the overall theme or argument?
  - Where applicable, are environmental descriptions vivid and effective at creating the desired atmosphere?
- **Language and Expression:**
  - Is the language accurate, precise, and grammatically correct?
  - Is the expression clear and unambiguous?
  - Does the writing style match the requirements of the prompt, genre, and intended audience?

## C.3 GENRE-SPECIFIC EVALUATION CRITERIA

To account for the diverse nature of writing, annotators also applied specific standards for each category. The following are representative examples.

### C.3.1 FICTION (E.G., SCI-FI, FANTASY, MYSTERY)

- **Characters:** Are characters consistent throughout the narrative? Are their relationships (e.g., friendship, rivalry) authentic and do they drive the plot?
- **Narrative Technique:** Does the author "show" the story through action, dialogue, and detail, rather than simply "telling" the reader what is happening?
- **Creativity:** Does the story demonstrate originality in its premise, characters, or plot? Does it effectively use narrative devices like foreshadowing and callbacks?

### C.3.2 SCRIPTWRITING

- **Dialogue:** Is the dialogue believable for the characters, reflecting their personality, background, and emotional state?
- **Action & Staging:** Does the script include stage directions (e.g., tone, emotion, action) for dialogue? Does it incorporate elements like set design, sound effects, and props?
- **Motivation:** Do the main characters have clear, understandable motivations that drive the plot forward?

### C.3.3 NON-FICTION (E.G., ESSAYS, BIOGRAPHIES, REVIEWS)

- **Accuracy:** Are all factual claims, data, quotes, and historical details accurate?
- **Authenticity:** Is the author's emotion and experience conveyed in a genuine and credible manner?

- **Depth:** Does the writing go beyond surface-level description to offer deeper analysis of causes, meanings, or connections?

### C.3.4 FUNCTIONAL DOCUMENTS (E.G., RESUMES, PROPOSALS, MEMOS)

- **Purpose:** Is the core purpose of the document (e.g., to inform, persuade, request) immediately clear?

- **Completeness:** Does the document include all information necessary to achieve its goal?

- **Format & Logic:** Does it follow the conventional format for its type? For persuasive documents, are the arguments clear, well-supported, and logical?

### C.3.5 FUNNY (E.G., INTERNET MEMES, COPYPASTA)

This category evaluates a model's grasp of niche, often non-literal communication styles.

- **Form:** Does the response deliberately break conventional logic for humorous or absurd effect?

- **Technique:** Does it correctly use techniques specific to the subculture, such as puns, homophones, context-dependent slang, or "serious nonsense"?

- **Tone:** Does it successfully capture a specific ironic or satirical tone, potentially with multiple layers of meaning?

## D DATASET STATISTICS

Table 3: Distributional properties of preference pairs in **WritingPreferenceBench**.

| Dataset | Metric | Type | Mean (STD) | Median |
|---------|--------|------|------------|--------|
| English | Length (words) | Chosen | 1450.3 (1801.9) | 961.5 |
| | | Rejected | 839.9 (593.4) | 792.0 |
| | Quality Score | Chosen | 2.913 (0.296) | 3.000 |
| | | Rejected | 1.602 (0.553) | 2.000 |
| Chinese | Length (words) | Chosen | 1873.5 (1967.5) | 1340.5 |
| | | Rejected | 1458.3 (1311.0) | 1218.5 |
| | Quality Score | Chosen | 2.560 (0.589) | 3.000 |
| | | Rejected | 1.115 (0.567) | 1.000 |

## E EXAMPLES OF BENCHMARK QUERIES

To illustrate the nature of the tasks in WritingPreferenceBench, this section provides several examples of the expert-crafted queries given to the models. These queries are designed to be specific, evocative, and challenging, pushing models beyond generic text generation.

### E.1 EXAMPLE 1: POETRY

**Query:**

> Please help me write a modern poem on the theme of the old refrigerator in my grandmother's kitchen. It no longer cools and is now used as a storage cabinet; there are faded stickers and an old shopping list still on its door. The poem needs to start with sensory details like its sound, its smell, and its appearance. It should be depicted as a guardian of family memories. Please use rhetorical devices like personification or metaphor to express a sense of nostalgia and affection for the old days.

### E.2 EXAMPLE 2: PRODUCT REVIEW

**Query:**

> Write a professional product review for a high-end outdoor shell jacket. The article must be well-structured and centered on its core performance metrics. The review should include at least: 1. Design & Workmanship: Analyze the jacket's fit and cut, fabric technology, seam sealing process, zipper configuration, and overall weight. 2. Core Functionality Test: Objectively evaluate its waterproofing, breathability, and windproofing performance in simulated or real-world conditions. 3. Details & Usability: Review the hood's range of adjustment, the logic of the pocket layout, and the adjustment systems for the cuffs and hem. The article's conclusion must clearly summarize the product's pros and cons and, in conjunction with its price, provide clear purchasing advice and an analysis of suitable user groups.

### E.3 EXAMPLE 3: FUNNY

**Category:** `ACGN Abstract Literature / Doujin (Fan Fiction)`

**Query:**

> Write an abstract fanfiction piece set against the backdrop of the "Human Instrumentality Project" from Neon Genesis Evangelion. The "protagonist" of this piece is not a specific person, but the very moment in which all human consciousnesses dissolve, merge, and collide within the Sea of LCL. The core theme is "the boundary between the Self and the Other," aiming to explore which is more terrifying: absolute loneliness or the loss of self through fusion. You do not need to construct a plot. Instead, use a fragmented, multi-vocal "stream-of-consciousness" style to weave together the internal monologues, memory fragments, and sensory perceptions of different characters—Shinji's inferiority complex, Asuka's arrogance, Rei's emptiness—and iconic sensory details (like "the metallic taste of orange juice" or "the sweetness of watermelon") to form a chaotic yet harmonious sea of consciousness.

### E.4 EXAMPLE 4: SHORT STORY

**Query:**

> Write a short story about a conflict between neighbors in a city. The protagonist is a young person who works from home and is constantly bothered by a strange, rhythmic noise coming from their new upstairs neighbor late at night. At the moment they can't stand it anymore and decide to confront the neighbor, they discover an unexpected and poignant truth about the source of the noise. The core of the story is the dramatic turn caused by this revelation, aiming to explore the alienation, misunderstanding, and eventual reconciliation between people in a modern city.

### E.5 EXAMPLE 5: ARGUMENTATIVE ESSAY

**Query:**

> On our journey through life, we often face the choice between "looking back" and "moving forward." Some believe that dwelling on the past hinders progress, and thus one must resolutely "move forward." Others are convinced that by frequently "looking back" and drawing wisdom from past experiences and lessons, we can walk the future path more steadily. These two attitudes, seemingly contradictory, are in fact dialectically unified, jointly shaping the trajectory of our lives. Please write an argumentative essay titled "'Looking Back' and 'Moving

Forward'". Your viewpoint should be distinct and your essay well-structured. The article must not only discuss why we should "look back" and why we must "move forward," but more importantly, it must delve into how a balance and unity can be achieved between the two. You are required to cite at least one historical figure as positive or negative evidence and analyze it in conjunction with a contemporary social phenomenon or a personal experience. When narrating the case, you must include rich descriptive details that reflect the character's internal journey and emotional changes when faced with the choice between "looking" and "moving." The essay should be no less than 800 words.

### E.6   EXAMPLE 6: SPEECH

**Query:**

You are about to graduate after three years of high school and, as the student representative, you need to deliver a speech at the graduation ceremony. Your audience includes not only the classmates you've spent every day with, but also the hardworking teachers and the parents who have come to attend. Please write a speech manuscript centered on the theme of "Gratitude and Responsibility." The speech should not be a collection of empty slogans or a simple farewell. You must include two specific, detailed stories: first, recount the profound impact a particular teacher had on you, describing a teaching moment or interaction that you still remember vividly; second, share a story of friendship and mutual growth with your classmates. Please create your own title. The speech must be no less than 600 words, with a closing signed by "Graduate Representative, Wang Chen" and the date.

### E.7   EXAMPLE 7: ADVERTISING COPY

**Category:** `Advertisement Copy / Marketing Email`

**Query:**

Write a marketing email for the new "Pathfinder 30L" backpack from the outdoor brand "Nomad's Gear." The email must use a vivid user story (e.g., a summit experience) to highlight the backpack's benefits, such as being lightweight and durable, in order to spark the reader's desire for adventure. A catchy subject line, an immersive story, and a clear call to action at the end are required.

### E.8   EXAMPLE 8: BIOGRAPHY

**Query:**

Please help me write a biography of the Mexican painter Frida Kahlo, with a suggested title of "The Burning Thorn Bush: Frida's Pain and Creation." The core theme of this biography should not be a simple chronological account of her life, but an in-depth exploration of "how pain became the core fuel for her artistic creation." You need to closely link and analyze key events in her life (such as the bus accident, her marriage to Diego Rivera, and her miscarriages) with her representative paintings. Provide a detailed interpretation of how she transformed physical disability and emotional turmoil into the powerful symbols and visual language of her artwork.

# F USE OF LLMs

