# OpenReview forum: "Beyond Correctness: Evaluating Subjective Writing Preferences Across Cultures"
_ICLR.cc/2026/Conference — Submitted to ICLR 2026_

### Official Review · Reviewer_GkYR · 2025-10-15

**Soundness:** 2
**Presentation:** 3
**Contribution:** 2
**Rating:** 2
**Confidence:** 4

**Summary:**

This paper presents a benchmark to assess how well different preference labelling approaches are able to evaluated based on "subjective" versus "objective" criteria. The "subjective" criteria primarily pertain to style and degree of creativity in a response, whereas "objective" cover attributes like grammatical correctness, coherency, and factual correctness. In the paper, the benchmark is used to assess 20 different LLMs falling into one of three categories: regression-head RMs, generative RMs, and LLM-as-a-Judge prompting. The LLMs are assessed both on English and Chinese preference pairs. The main findings are that the LLMs are bad at assigning preference labels based on "subjective" criteria, with generate RMs performing best. A performance discrepancy is observed between the English and Chinese samples, with better performance on the English samples, and across creative writing task macro-categories. The main conclusion of the paper is that current LLMs are brittle at the preference labelling task, and intermediary reasoning steps are beneficial.

**Strengths:**

- Although the conclusions about preference labelling with LLMs is highlighted in the paper, it presents a benchmark to assess preference labelling quality when quality factors such as grammatical and factual correctness and coherency are held constant between the responses in a given preference pair. This essentially removes the lower hanging fruit from the preference evaluation.
- Humans were involved in all stages to create and curate the dataset. From the expert/annotator descriptions, it sounds like they should have had appropriate skills levels for that tasks they completed.
- Many LLMs across three different categories of preference labelling strategy were evaluated.
- Multiple creative writing domains were assessed.
- The responses included in the benchmark come from multiple LLMs, not a single LLM.
- The paper is well written and easy to read/follow.

**Weaknesses:**

- The paper is framed to highlight the learnings from evaluating the 20 LLMs on the proposed benchmark. However, the main learnings either primarily replicate findings that have been previously established in the literature (e.g. reward models can be brittle -- https://arxiv.org/pdf/2210.10760 and English vs. non-English RM quality -- https://arxiv.org/pdf/2410.15522), are general understandings (e.g. specialized models perform better on the target task than generalist models and models perform worse in OOD settings), or do not appropriately address the impact distributional differences in training versus evaluation data by focusing on the need to improve training objectives without considering the issue may be related to training data issues (e.g. the performance of regression-head RMs, which have been trained on a specific preference objective that likely differ's from the benchmark's preference objective). I think it is completely appropriate to introduce a benchmark such that results one the benchmark replicate what has been previously demonstrated across multiple papers and studies, as this allows the benchmark to measure progress on these issues. However, the paper must be framed in this way instead of presenting these types of findings as new. Additionally, it is great to have a benchmark able to assess LLM performance on a task they are not typically explicitly trained to performed, however this should be recognized in the discussion/conclusion instead of pointing immediately to issues with training objectives. For example, most RMs on not trained on a lot of creative writing tasks (something the paper calls out) and most public annotator protocols do not instruct the annotators to prioritize creative responses.
- With the details currently given, I could not replicate this paper (neither the benchmark nor the results).
- The benchmark target "subjective" quality dimensions, which seems to mostly mean creative responses. The paper presents many results about how well LLMs are able to achieve this task. However, this is not all together surprising as subjective differences by their very nature are more difficult to replicate assessments of both with people and models. The data is annotated by only 3 annotators per sample and the paper describes what sounds like an intensive calibration process among annotators. This leaves open the question, how "universal" versus annotator-specific are these subjective preference labels? For example, if a different group of 3 annotators came in to the complete the task, how well would they replicate the labels of the original annotators?
- LLM performance on this benchmark should be contextualized with both RewardBench performance and multi-lingual abilities (potentially  using https://arxiv.org/abs/2410.15522).This will quantify how much of a RM performance drop occurs when assessing quality along dimensions relevant to creative writing.
- The LLM prompts are not provided. This is important as the instructions the LLM receives will obviously have a large impact on preference labelling performance. For example, if provided instructions about the importance of creative writing attributes, does accuracy improve?
- The introduction and conclusion make general claims about scale not impacting performance. These need to amended to be specific to the LLM judge results.
- It looks like there is a mismatch in the numbers provided per macro-category in the text (i.e. lines 236/237) versus in Figure 3.
- The connection between response length and the claim about "...mediocrity converges toward formulaic patterns, creativity manifests across diverse scale[,]" is not clear.
- Generative RMs without reasoning before scoring are missing, which limits the strength of the take away that it is the reasoning change versus generative RM that leads to improved performance over the regression-head RMs. This is an especially important comparison given that reasoning steps were not found to be beneficial for the LLM judges.

**Questions:**

- Was there any correlation between overall LLM judge prompt length and preference label correctness? It looks like many of the chosen and rejected responses were at least 1000 tokens long. If the prompts were too long, it makes sense the performance of the LLM judges would be limited.
- Please provide citations for the claim, "...prevailing assumption that enhanced reasoning capabilities, such as chain-of-thought, naturally lead to better alignment with complex human values." Is it that LLM judges with advanced reasoning is thought to be better? Or that these are easier to set up for an arbitrary preference labelling task than to collect preference data and train a RM?
- How diverse are the queries across macro-categories? Specifically in the data that was used. How many unique queries were retained after the filtering processes were applied?
- How many responses per each of the "20 state-of-the-art models" used to generate the diverse responses were retained in the final version of the benchmark?
- How were the "traditionally underrepresented genres" (line 240) identified and selected?
- Were there 7 annotators per language? Or 7 total? If 7 total, how many per language?
- There is a lot of emphasis on creativity. How was this defined/operationalized?
- What was the inter-annotator agreement? How exactly were the yes/no questions in Appendix C.2 and C.3 used? You talk about directional agreement on line 179. Is the pivot point for the direction >2 and <=2?
- Point 3. (line 181 - 182), what was the complete set of "confounding factors"?
- What is a blueprint? What are some examples? What does it mean for a blueprint to be expanded upon? What was the prompt that was used? Please provide example blueprints and queries.
- How were "grammatical errors, factual inconsistencies, and clear prompt violations" detected? How performant was this filtering process? What are examples of data that was removed?

---

### Official Review · Reviewer_k7ey · 2025-10-28

**Soundness:** 3
**Presentation:** 3
**Contribution:** 2
**Rating:** 6
**Confidence:** 4

**Summary:**

The work introduces the WritingPreferenceBench, an expert annotated dataset for writing preference in English and Chinese across different writing domains. Experiments for automatic evaluation assessment through reward models or base LLMs are then conducted, showing that generative reward models outperform non-generative ones (classifiers), and also outperform out-of-the-box LLMs performing LLM-as-a-judge through prompting.

**Strengths:**

- The paper is well written, and the communication of contributions is fairly clear.
- The annotated dataset seems like it would valuable to the community, and spans two languages and 8 domains. It is of decent size when taking into account the difficulty of annotation in the creative writing space.
- The results are thorough, evaluating a wide set of models, both general purpose and specialized reward models.

**Weaknesses:**

- The annotated data is the principal contribution of the work, yet several key details are not described in enough detail to give confidence about the results. Here are some questions that would require clarification to strengthen the contribution:
1. The unit for response length is not defined, are these characters, tokens, words? (I assume characters but unclear)
2. Some of the judged responses are quite long (6k "units"), which would require a long duration to read and judge. Were annotators compensated fairly and did they have enough time to read samples before making judgements. How were annotators recruited, compensated, and what were the guidelines around how much time they should spend per sample?
3. The final pair curation guidelines seem good, but they could also introduce some bias in the dataset. What is the level of agreement that occurred on the non-filtered dataset? What is the percentage of the annotated data that did not pass the validation criteria.

**Questions:**

1) See the questions listed above in the Weaknesses section.
2) Regarding the use of generative reward models, since they seem to work the best on your dataset, have you analyzed the chain-of-thought/reasoning produced by these models, and what are the the key patterns observed within those that could explain the gains in performance. Do they align with the guidelines you provided the trained experts?
3) The Writing Quality Reward Model (https://huggingface.co/Salesforce/WQRM-PRE) might be worth exploring in your analysis, as it seems related in that it is a specialized models for writing quality assessment.

---

### Official Review · Reviewer_58Yw · 2025-10-29

**Soundness:** 3
**Presentation:** 3
**Contribution:** 2
**Rating:** 2
**Confidence:** 4

**Summary:**

This paper introduces a benchmark consisting of 1800 preference judgments for a variety of creative writing tasks in English and Chinese. The authors test various reward models (sequence-based, 0-shot LM, using reasoning) and find that all except for those using explicit reasoning do not match up with human judgments. Because the construction of the benchmark consisted of controlling for obvious, objective errors, the authors suggest that this low performance is because of reward models being trained to spot objective errors rather than robustly learn notions of quality, especially for more subjective tasks.

**Strengths:**

1. The benchmark produced will likely be a valuable resource for developers primarily serving English/Chinese users.
2. The comparison across various classes of reward models is thorough.
3. To my knowledge, the process of disentangling objective factors (e.g. clear mechanical issues) with subjective factors is novel and very interesting.
4. For the most part, the pipeline for constructing the dataset is thorough and robust.

**Weaknesses:**

1. I think the benchmark could be more valuable if it either (or both) included more languages or more tasks. While the former would make validation a lot more difficult, I think the latter could at least be used to expand to tasks that are somewhat less subjective than creative writing.
2. I would have liked to have seen more analysis regarding inter-annotator agreement. While it is mentioned that part of the statistical validation included filtering for pairs where there is at least some directional consensus, I think more analysis would be helpful since these tasks are so subjective.
3. I think that the conclusion that models "lack the underlying framework to encode and weigh aesthetic qualities like
originality, emotional resonance, or stylistic flair" is a bit too strong in this context. If certain tasks are subjective, shouldn't it be expected that reward model accuracy should be closer to 50%? Moreover, if the tasks are subjective, is the assumption of having a ground truth at all too strong? I worry that assuming this and using reward model's trained on this benchmark may significantly reduce the creative ability of the resulting aligned model.

**Questions:**

1. Given that some pairs were filtered out for not having certain properties (i.e. directional agreement) but "objective factors" are controlled for, do you have a sense of what some of the factors were for differentiating the final set of preferred/dis-preferred responses? Could there have been some biases prevalent here?
2. Related to 1, does the filtering step result in pairs corresponding to prompts that are in some sense more objective than others?
3. Did you do any experiments with the pairs that were not used due to lack of consensus?

---

### Official Review · Reviewer_QSfx · 2025-10-30

**Soundness:** 2
**Presentation:** 3
**Contribution:** 2
**Rating:** 6
**Confidence:** 5

**Summary:**

This paper introduces WritingPreferenceBench, a cross-lingual benchmark of 1,800 human-annotated preference pairs (1,200 English, 600 Chinese) spanning 8 creative writing genres. Each pair is controlled for grammatical correctness, factual accuracy, and length to isolate subjective writing quality. Through analysing the performance across 21 models, there are three key observations:
1. General-purpose language models underperform reward models
2. Reasoning capabilities alone do not ensure better judgment
3. LLMs exhibit strong biases and instability across genres, languages, and evaluation settings

Overall, the paper argues that current RLHF and LLM-as-judge paradigms optimize for objective error detection rather than subjective quality assessment, and that intermediate reasoning representations may be essential for modeling human aesthetic preferences.

**Strengths:**

This paper propose a rigorously aggregated benchmark that cleanly assess LLMs' objective writing preferences. Comprehensive writing and holistic experimental design provide strong support for the findings observed by the author. Several points that distinguish this paper can be clearly outlined.
- Well drafted paper and demonstrated work target on clear object
- Comprehensive description regarding the benchmark collection, experimental setting, and results analysis
- The experiments are standardized, covering 21 models with detailed genre
- Overall, the work makes a meaningful contribution by providing actionable insights into the architectural requirements for modeling human creative preference

**Weaknesses:**

- In line 162, the author mentioned they recruited 7 expert annotators with a calibration phase to align their standard of creativity. I don't find any metric that measure the correlation between experts before and after calibration. This could be fatal if the calibration phase fail to make an agreement between annotators
- From Figure 4, there is a noticeable difference between the distributions of chosen and rejected sentences. However, the author claims that the benchmark isolates the subjective factor. My concern is that this mismatch may not provide sufficient support for the claim
- In table 1, the results are highlighted by different colors when its over 70% or lower than 50%. But there is no justification on how the value was decided

**Questions:**

Q1: This paper devotes significant effort to describing a clear pipeline for data aggregation, but it lacks an analysis of the essential factors underlying "creative writing."

Q2: What conclusions can be drawn based on the performance gap between the reward model and general large language models (LLMs)?

---

### Meta-Review · Area_Chair_Jc6W · 2026-01-08

**Summary:**

The paper introduces WritingPreferenceBench, 1,800 human preference pairs across English/Chinese and 8 creative-writing genres, and finds most standard reward models / LLM-judges are near chance, while generative reward models with explicit reasoning perform substantially better and show large genre variance. The reviewer comments are split, with the concerns on missing details and reproducibility, and the authors did not provide rebuttal.

**Reviewer Concerns:**

There is no rebuttal provided.

**Reviewer Scores:**

There is no rebuttal provided.

---

### Decision · Program_Chairs · 2026-01-26

Reject